# Semantic Alignment for Multimodal Large Language Models

## ABSTRACT

Research on **M**ulti-modal **L**arge **L**anguage **M**odel**s** (**MLLMs**) towards the multi-image cross-modal instruction has received increasing attention and made significant progress, particularly in scenarios involving closely resembling images (*e.g.*, change captioning). Existing MLLMs typically follow a two-step process in their pipelines: first, extracting visual tokens independently for each input image, and then aligning these visual tokens from different images with the Large Language Model (LLM) in its textual feature space. However, the independent extraction of visual tokens for each image may result in different semantics being prioritized for different images in the first step, leading to a lack of preservation of linking information among images for subsequent LLM analysis. This issue becomes more serious in scenarios where significant variations exist among the images (*e.g.*, visual storytelling). To address this challenge, we introduce **S**emantic **A**lignment for **M**ulti-modal large language models (**SAM**). By involving the bidirectional semantic guidance between different images in the visual-token extraction process, SAM aims to enhance the preservation of linking information for coherent analysis and align the semantics of different images before feeding them into LLM. As the test bed, we propose a large-scale dataset named **MmLINK** consisting of 69K samples. Different from most existing datasets for MLLMs fine-tuning, our MmLINK dataset comprises multi-modal instructions with significantly diverse images. Extensive experiments on the group captioning task and the storytelling task prove the effectiveness of our SAM model, surpassing the state-of-the-art methods by a large margin (+37% for group captioning and +22% for storytelling on CIDEr score). Project page: https://anonymous.4open.science/r/SAM-F596.

## CCS CONCEPTS

• **Computing methodologies → Artificial intelligence**.

## KEYWORDS

Multi-modal Large Language Models, Multi-image Reasoning, Semantic Alignment, Bidirectional Semantic Guidance Mechanism

## 1 INTRODUCTION

Multi-modal Large Language Models (MLLMs) [9, 23, 24, 28, 49, 50] have shown great potential in processing multi-image cross-modal instructions, with GPT-4V(ision) [32] standing out as a leading example. Upon reviewing the preliminary exploration report [47] of GPT-4V, we observe that its successful applications in handling

*ACM MM, 2024, Melbourne, Australia*
© 2024 Copyright held by the owner/author(s). Publication rights licensed to ACM.
ACM ISBN 978-x-xxxx-xxxx-x/YY/MM
https://doi.org/10.1145/nnnnnnn.nnnnnnn

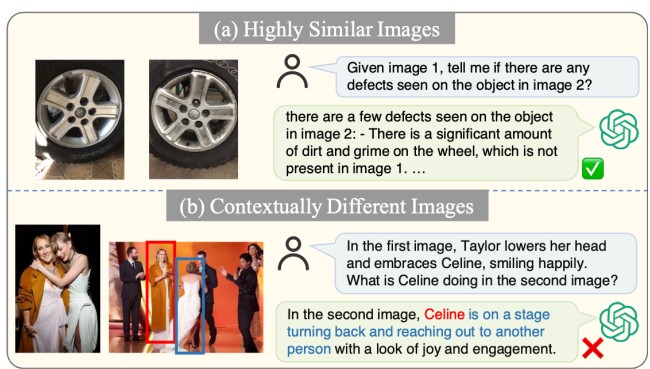

**Figure 1: GPT-4V shows great performance in pinpointing differences between highly similar images (a), but struggles to align character identities across images featuring notably varied character poses, camera angles and contexts (b). Example in (a) is sourced from [47].**

multi-image tasks primarily focus on closely resembling images, as shown in an example from the report (see Figure 1(a)). In such cases, where images share almost identical background contexts, MLLMs can effectively address the reasoning tasks by aligning the similarities and pinpointing the differences between the visual contents. However, the effectiveness of MLLMs, including GPT-4V, diminishes when faced with significantly different images in terms of content, context, or style, as illustrated in Figure 1(b). In such cases, MLLMs may struggle to establish meaningful semantic connections between the images and generate accurate responses.

We attribute the deficiency to the absence of semantic alignment in existing MLLMs when processing multiple images. To see this, we revisit the two stages in the processing pipeline of MLLMs for multi-image instructions:

- **Independent Perception.** This step employs a visual tokenizer (*e.g.*, Q-Former [23], linear projection [28] or the Resampler [1]) to map the visual features encoded by a pre-trained vision backbone (*e.g.* EVA-CLIP [10]) to visual tokens within the feature space of the foundation LLM. In this way, the projected visual tokens of an image carry its visual features.

- **Integrated Analysis.** After obtaining the projected visual tokens of each image, the model fuses them with the textual tokens of the instruction using cross-modal attention mechanisms [42]. This step allows the model to combine information from different modalities and images.

In this fashion, MLLMs mainly tackle multi-image reasoning in the integrated analysis step, which leads to semantic misalignment problems that hinder the discovery of inter-image correlations. This is because 1) the visual tokens serving as input to the integrated analysis step may lack crucial 'linking' information (*e.g.*, character's identity in Figure 1(b)) necessary for identifying correlations, especially when the input images exhibit diverse contexts.

Specifically, the visual tokens are influenced by the inductive bias from training data like image-caption pairs, which lead them to selectively express salient visual contents [12, 24]. Therefore, affected by complex context, the focus of visual tokens of different images may be different, hindering semantic alignment. 2) In the integrated analysis stage, LLMs perform attentions on visual tokens and textual tokens jointly, which leads to the semantic alignment between visual tokens being overwhelming in a large number of token interactions [7, 36].

Our solution to address the aforementioned semantic misalignment problem is to introduce contextual semantics from contextual images (*i.e.,* images other than the currently perceived image) as guidance in the perception stage. By utilizing contextual images as references, we may accurately align the extracted visual tokens of the currently perceived image with contextual semantics before the integrated analysis step. However, it is crucial to note that not all contextual images in the input multi-modal instruction may exhibit direct correlations with the currently perceived image due to irrelevant or even noisy information. In such cases, directly incorporating the whole information from all contextual images may pose challenges to the perception process. Therefore, in addition to the semantic misalignment problem mentioned above, how to accurately extract the contextual semantics highly relevant to the currently perceived image is another crucial problem deserving careful consideration.

To tackle the above issues, we propose the **S**emantic **A**lignment method for **M**ultimodal large language models (**SAM**) towards the multi-image cross-modal instruction. The SAM model tackles the misalignment problem between multiple images in the input multi-modal instruction by incorporating the bidirectional semantic guidance mechanism. Specifically, this mechanism involves two interacting steps: (1) SAM extracts visual tokens from the currently perceived image based on the natural language prompt, with the guidance of the contextual semantics from the contextual images (*i.e.,* the other images in the input multi-modal instruction). (2) To obtain representative contextual semantics from the contextual images, SAM incorporates a novel visual tokenizer called W-former, which is specifically designed to extract synchronous semantic information from multiple images. With the visual tokens from the currently perceived image, the W-former first employs an adaptive adjustment module at the patch level of each contextual image in the multi-modal instruction, to enhance the prominence of critical information. Subsequently, the W-former incorporates the aligned patch features and the language prompt to extract the contextual semantics for the visual-token perception from the currently perceived image.

Most existing multi-image datasets applied for fine-tuning the multi-modal large language models may demonstrate substantial similarities between images of the multi-modal instruction [24], lacking associated images with significant difference (*e.g.,* different contexts, variant styles). To this end, we introduce a novel large-scale multi-modal dataset named **MmLINK** for the MLLMs research. It contains 69K vision-language samples, specifically designed to enhance the model's abilities of cross-modal multi-image semantic alignment and correlation mining. Different images within each multi-modal sample of our dataset exhibit both semantic-level correlations and significant visual differences (*e.g.,* the same person

in different images with different poses and contexts). Through extensive experiments, we showcase that training on MmLINK leads to significant performance improvements in our SAM model, surpassing the current state-of-the-art models.

Overall, our contributions are summarized as follows:

- To the best of our knowledge, we embark on the early exploration of the semantic correlation between substantially diverse images in the multi-modal instruction, within the MLLMs field. As the research foundation, we construct a large-scale comprehensive dataset called MmLINK comprising 69K samples.
- We propose the multi-modal model named SAM, which employs bidirectional semantic guidance between images of the input multi-modal instruction to align the semantics of different images in the perception stage.
- Extensive experiments prove the effectiveness of our SAM model by significantly outperforming the state-of-the-arts.

## 2 RELATED WORK

### 2.1 Multi-modal Large Language Models

Multi-modal pretraining [8, 16, 18, 40, 45, 48] aims to train models on varied datasets across multiple modalities to capture cross-modal correlations. Building upon the concept of multi-modal pretraining, Multi-modal Large Language Models represent a fusion of multi-modal pretraining techniques with the advanced capabilities of Large Language Models. Flamingo [1] proposes architectural innovations that enable the integration of powerful pre-trained vision-only and language-only models, managing sequences of interleaved visual and textual data, and seamlessly ingesting images or videos as inputs. BLIP2 [23] employs a lightweight Querying Transformer (Q-Former) that bridges the modality gap through a two-stage pre-training process. Following closely are LLaVA [28], MiniGPT-4 [50] and InstructBLIP [9], which conduct instruction tuning [33] on MLLMs. They utilize different modules to bridge the gap between image modality and text modality, like Q-Former used in InstructBLIP and an linear layer employed in LLaVA. Their training data includes a wide range of single images and textual question-answer pairs, enhancing model's instruction-following ability. Cheetah [24] and MMICL [49] extend their work into the field of multi-image reasoning, featuring a visual tokenizer for visual token extraction and leverage LLMs for integrated analysis. However, almost all these MLLMs tackle multi-image understanding in the integrated analysis step, leading to ignorance of contextual semantics in the perception step and absence of crucial 'linking' information necessary for identifying correlations. To tackle this, we propose a bidirectional semantic guidance mechanism that incorporates contextual semantics during visual token extraction, enhancing multi-image correlation alignment.

### 2.2 Multi-image Captioning

Multi-image captioning [6, 14, 31, 41] is an advanced domain within the field of multi-modal research that focuses on generating descriptive text for a set of images, rather than just a single image. This area extends the capabilities of traditional image captioning by considering multiple images simultaneously, aiming to produce

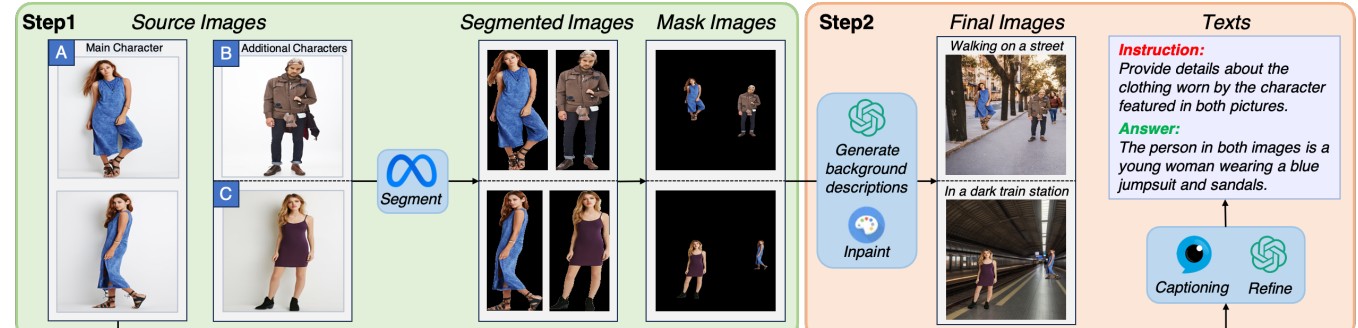

**Figure 2: Demonstration of our proposed 2-step sample synthesis pipeline. We begin by selecting images featuring characters in different poses (A), along with 2 another distinct characters (B, C). The selected images are segmented to isolate each character, after which they are merged into mask images. Inpainting technology is then utilized to fill in the background areas of these mask images to obtain the final images, using descriptions generated by ChatGPT. Text annotations are generated by InstructBLIP and further refined with ChatGPT.**

coherent and contextually relevant captions that encapsulate the collective content and narrative of the images. Significant advancements have been made in multi-image captioning tasks, such as group captioning [5, 25] and visual storytelling [4, 15, 27, 44, 46]. The challenge of multi-image captioning lies in its requirement for a deeper understanding of the relationships between images, the story they tell together, and how elements from each image relate to one another. Existing models have introduced methods to explore inter-image correlations. For instance, Li et al. [25] fuse visual features from multiple images using self-attention [42] mechanism. Liu and Keller [27] focus on the importance of characters in visual storytelling and propose two task: important character detection and character grounding in visual stories, and develop unsupervised models for these tasks using distributional similarity and pre-trained vision-language models. However, these proposed fusion methods usually require training the entire model, which will lead to significant training costs for large language models. Therefore, we propose a more precise structural design that achieves semantic alignment by introducing fewer trainable parameters.

## 3 TRAINING DATA CURATION

Most training datasets for Multi-modal Large Language Models (MLLMs) primarily focus on single-image multi-modal instructions. However, the development of multi-modal datasets that incorporate correlations across multiple images for training is still in its early stages of exploration. The current methods for constructing multi-image multi-modal datasets mainly focus on creating highly similar images for correlation establishment. For instance, some researchers [19, 24] modify minor details of the original image while leaving the majority unchanged, to generate other images in the multi-image multi-modal instruction. However, these construction methods overlook the crucial 'linking' information across diverse contexts (*e.g., the same person in different images with different actions and poses*), which are often more significant for training the semantic alignment ability of MLLMs.

Toward this issue, we introduce a novel 2-step sample synthesis pipeline as shown in Figure 2 to construct our multi-modal dataset,

**MmLINK**. In the first step, we curate different images featuring the same object but with varying poses, lighting conditions, or view angles (*e.g.*, the same person with different actions) to create the image groups, where each group corresponds to one object. We then segment the shared object from the images within each image group to obtain our segmented objects and combine these segmented objects to generate mask images. In the second step, we inpaint the mask images based on descriptions generated by LLM to create a large number of semantically correlated images with different contexts. This approach ensures that the dataset includes diverse correlations across multiple images, enhancing the training process for MLLMs. In order to create the textual components of a training sample, we automatically generate a language query and its corresponding answer for two images depicting the same object but in different contexts.

**2-step Sample Synthesis Pipeline.** Each sample in our MmLINK dataset consists of an image pair, a corresponding language instruction, and a textual answer for the instruction. The images in the same pair depict the same object with different poses, lighting conditions, or view angles. To ensure the diversity of the image objects in our MmLINK dataset, we collect the images with variance objects (*e.g.*, characters, furniture items, icons, and book covers) from different image sources. Notably, for different types of objects, slightly various pipelines are employed to create the corresponding samples. We take the 'character' as an example to illustrate the sample construction pipeline, as depicted in Figure 2. In addition, we provide information on the construction pipeline for samples involving other objects in supplementary Section 1.1.

In the first step, we construct image groups for different characters from the DeepFashion dataset [29], in which the same character exhibit identical clothing but different poses and view angles. We select two images of the same character with varied poses as the main character (*e.g.*, A in Figure 2), along with two another distinct characters as additional characters (*e.g.*, B and C in Figure 2). Next, we segment each character from the source image using Segment Anything [20]. We combine the main character with each of the two additional characters and place them onto a $512 \times 512$ mask

image with diverse positions and sizes. In this way, we obtain two mask images featuring the same character, *i.e.*, the main character.

In the second step, we inpaint the two mask images with diverse background descriptions generated by ChatGPT[1]. This is achieved by Stable-Diffusion-Inpainting [39], which can generate background pixels smoothly integrated with the characters according to the background descriptions. The two inpainted images serve as the visual components of each training sample in MmLINK, challenging models to identify the same character across images despite the varied contexts and presence of additional characters. Finally, we employ InstructBLIP [9] to generate descriptions of the main character and use ChatGPT to obtain refined descriptions, which serve as the textual components in the final multi-modal samples.

**Quality Control.** To guarantee the quality of our dataset, we filter out the noise samples generated in the image-segmentation and image-inpainting steps in our pipeline. Specifically, during the image-segmentation process, we utilize CLIP [37] to calculate the similarity between the segmented characters and their corresponding character descriptions generated by InstructBLIP. This process serves to filter out incompletely segmented characters with a similarity score below 0.8. During the image-inpainting process, the CLIP model is applied to calculate the similarity score between synthetic images and their corresponding background descriptions, and filter out the inpainted images which mismatch their corresponding background descriptions with scores below 0.8.

## 4 METHOD

In this section, we will introduce our proposed **S**emantic **A**lignment method for **M**ultimodal large language models (**SAM**) in detail.

### 4.1 Overview

As illustrated in Figure 3, the pipeline of SAM can be formulated in three steps. Firstly, given $N$ input images, the vision encoder transforms them into patch-level features. For the $i$-th image, we represent $P$ corresponding patch-level features with $\mathcal{I}_i = \{\mathcal{I}_{i,j}\}_{j=1}^{P}$, where $P$ is the patch number and $j$ denotes the patch index. Then, we utilize the *Bidirectional Semantic Guidance* mechanism in the image perception stage to generate the visual tokens from the patch-level features with enhanced semantic alignment. Finally, in the integrated analysis stage, the large language model processes the visual tokens together with the input textual query and generates final prediction. In the following sections, we use subscript $i$ to indicate symbols relevant to the currently perceived image.

**Bidirectional Semantic Guidance.** Towards the semantic misalignment problem in existing MLLMs when processing contextually different images, we introduce the bidirectional semantic guidance mechanism in the image perception stage. It comprises two interactive processes, including *Part A: Assisted Visual Token Extraction* and *Part B: Contextual Semantic Generation*. Initially, in Part A, the Q-former layers are applied to process the currently perceived image based on the natural language query and extract the initial visual tokens $\mathbf{h}_i$. Next, in Part B, we propose a novel visual tokenizer called W-former. Guided by initial visual tokens $\mathbf{h}_i$, W-former extracts synchronous contextual semantics $\mathbf{c}_i$ from

[1]https://openai.com/blog/chatgpt. The prompts of ChatGPT used for dataset construction are provided in supplementary Section 1.2.

contextual images (*i.e.*, images other than the currently perceived image). Finally, the contextual semantics $\mathbf{c}_i$ are passed back to the Q-former layers in Part A to guide the updating of visual tokens by aligning them with the contextual information based on the natural language query.

### 4.2 Part A: Assisted Visual Token Extraction

Most existing Multi-modal Large Language Models (MLLMs) often generate visual tokens for each input image in the multi-image cross-modal instruction independently. This may lead to the absence of semantic alignment between images with diverse contexts, thereby hindering the ability to identify inter-image correlations. To address this issue, we introduce contextual semantics from contextual images of the multi-modal instruction and the natural language query to guide the extraction of visual tokens for the currently perceived image. These contextual semantics are provided from Part B, which also receives initial visual tokens of currently perceived image generated in Part A as guidance, thereby forming an interactive mechanism between the two processes.

Specifically, Part A is implemented with the transformer-based visual tokenizer, Q-former [9]. Q-former consists of several cross-attention layers, which concatenates pre-trained query vectors and textual tokens as the Query and the current image features $\mathcal{I}_i$ as the Key and Value, allowing them to interact with each other to obtain final visual tokens. We define the initial visual tokens $\mathbf{h}_i$ as the inputs at the $l$-th intermediate Q-former layer $\mathbf{q}_i^l$:

$$\mathbf{h}_i := \mathbf{q}_i^l \tag{1}$$

The initial visual tokens will serve as guidance to help Part B generate contextual semantics strongly related to the currently perceived image, by effectively reducing redundant details of contextual images in the multi-modal instruction.

After Part B generating contextual semantics $\mathbf{c}_i$ (see Equation 6), we select a certain subsequent layer $k$ ($k \geq l$) and add $\mathbf{c}_i$ to the input queries of the $k$-th Q-former layer $\mathbf{q}_i^k$ as a guidance:

$$\bar{\mathbf{q}}_i^k = \mathbf{q}_i^k + \mathbf{c}_i \tag{2}$$

where $\bar{\mathbf{q}}_i^k$ denotes the updated input queries. The subsequent Q-former layers will refer to the contextual semantics to generate the final visual tokens for the currently perceived image with enhanced semantic alignment.

### 4.3 Part B: Contextual Semantic Generation

The contextual semantics $\mathbf{c}_i$ from contextual images of the multi-modal instruction are needed to guide the extraction of visual tokens for the currently perceived image in Part A. However, it is important to note that not all contextual images in the input may have a direct correlation with the currently perceived image and the natural language query. Furthermore, even highly related images may still contain irrelevant or noisy information. To tackle these issues, we involve the W-former module to Part B. W-former contains Q-former layers which share the same parameters with the one in Part A, along with an novel adaptive adjustment design.

**Adaptive Weights.** Specifically, we denote the contextual image set (*i.e.*, images other than the currently perceived image) as $\{\mathcal{I}_m\}_{m \neq i}$. The patch-level features of the $m$-th ($m \neq i$) image are

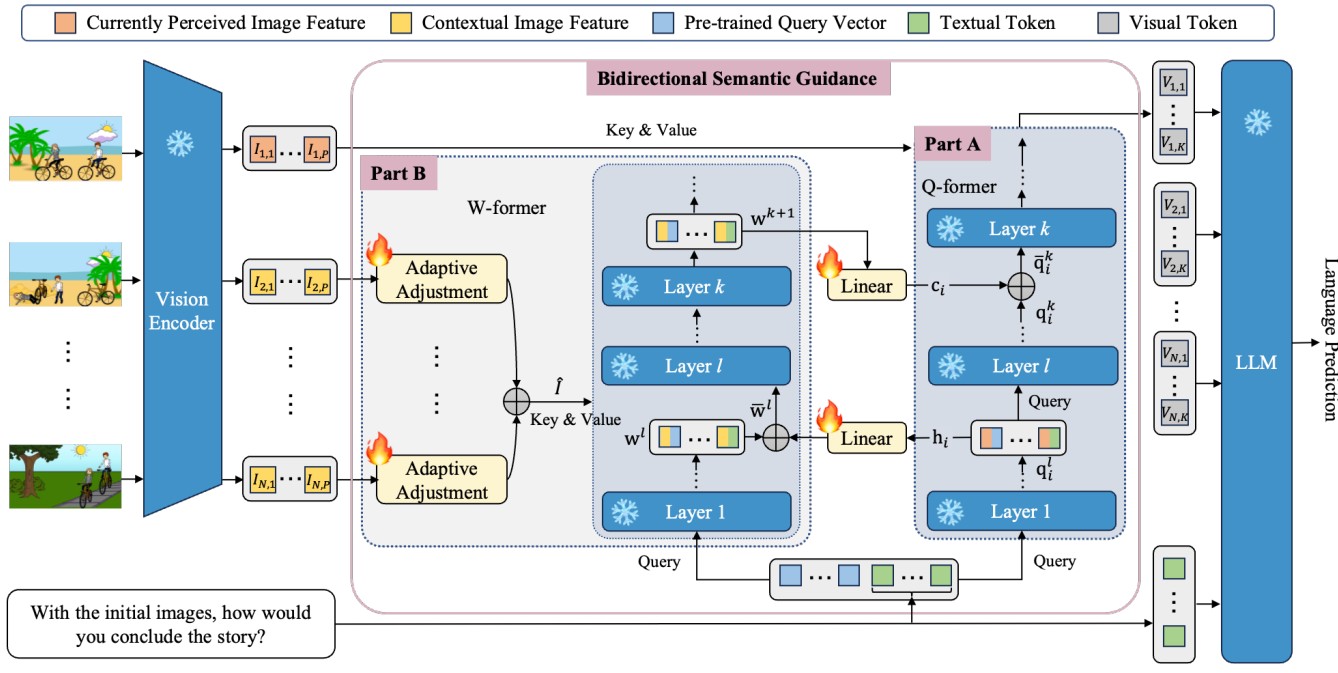

**Figure 3: Overview of SAM.** The core mechanism of our SAM model is the Bidirectional Semantic Guidance mechanism with two interactive processes: Assisted Visual Token Extraction (Part A) and Contextual Semantic Generation (Part B). In Part A, the Q-former module leverages the contextual semantics $c_i$, which are generated from contextual images (*i.e.*, images other than the currently perceived image) in the multi-modal instruction in Part B, to guide the extraction of visual tokens from the currently perceived image features. In Part B, the W-former module is utilized to select the contextual semantics from the visual context of contextual images. This selection process is facilitated by the attention mechanism in the adaptive adjustment, along with assistance from the initial visual tokens $h_i$, which are extracted from the currently perceived image in Part A.

denoted as $\{\mathcal{I}_{m,j}\}_{j=1}^P$. Then, we assign weights to each image patch with our proposed *Adaptive Adjustment* module, eliminating the prominence of irrelevant patches. Firstly, the adaptive adjustment module uses the linear layer **Linear**$(\cdot)$ and the softmax function **softmax**$(\cdot)$ to generate normalized patch-level weights $\bar{\alpha}_{m,j}$:

$$\bar{\alpha}_{m,j} = \textbf{softmax}(\textbf{Linear}(\mathcal{I}_{m,j})) \qquad (3)$$

Then, the adaptive adjustment module re-weights each contextual patch features and sums up the re-weighted patch features to obtain the merged contextual features $\hat{\mathcal{I}}$:

$$\hat{\mathcal{I}} = \{\hat{\mathcal{I}}_j\}_{j=1}^P, \quad \hat{\mathcal{I}}_j = \sum_{m \neq i} (\bar{\alpha}_{m,j} * \mathcal{I}_{m,j}) \qquad (4)$$

This process reduces the influence of irrelevant patches and amplifies important ones, improving patch-level alignment between the merged contextual features $\hat{\mathcal{I}}$ and the currently perceived ones. The detailed framework of the adaptive adjustment module is illustrated in supplementary Section 2.

**Contextual Semantics.** After obtaining merged contextual features $\hat{\mathcal{I}}$, the W-former module generates the contextual semantics $c_i$. To enhance the specificity of $c_i$ with respect to the currently perceived image, we introduce its initial visual tokens $h_i$ (see Equation 1), defined as the input queries of the $l$-th Q-former layer in Part A, as a guiding factor. This is achieved by incorporating the initial visual tokens with the input queries of the corresponding

$l$-th W-former layer via a linear layer **Linear**$(\cdot)$:

$$\bar{\textbf{w}}^l = \textbf{w}^l + \textbf{Linear}(\textbf{h}_i) \qquad (5)$$

where $\textbf{w}^l$ denotes the original input queries of the $l$-th W-former layer and $\bar{\textbf{w}}^l$ denotes the updated ones. Guided by the initial visual tokens of currently perceived image, the remaining W-former layers could efficiently extract relevant details from merged contextual features $\hat{\mathcal{I}}$. At the corresponding $k$-th ($k \geq l$) layer, we derive contextual semantics from the output tokens $\textbf{w}^{k+1}$ using a linear layer **Linear**$(\cdot)$:

$$\textbf{c}_i = \textbf{Linear}(\textbf{w}^{k+1}) \qquad (6)$$

The resulting $\textbf{c}_i$ aggregates pertinent information to the currently perceived image from the entire context, facilitating the generation of visual tokens aligned with contextual semantics.

## 4.4 Efficient Training

We perform both interactions at the final layer of the Q-former and W-former, which means $l = k = 12$ (defined in Section 4.2). We compare the different choices of $l$ and $k$ in Section 5.3. For efficient training, we freeze the parameters of the vision encoder and the LLM. In addition, the Q-former layers are also frozen. This means that only three linear layers and the LLM projection layer require fine-tuning, constituting just **4.3M (0.05%)** trainable parameters.

**Table 1: Performance on group captioning and storytelling tasks. The best is bolded and the second best is underlined.**

(a) Group Captioning

| Model | Conceptual | | | Animal | | | Vehicle | | | Average | | |
|---|---|---|---|---|---|---|---|---|---|---|---|---|
| | ROUGE-L | CIDEr | BLEU-4 | ROUGE-L | CIDEr | BLEU-4 | ROUGE-L | CIDEr | BLEU-4 | ROUGE-L | CIDEr | BLEU-4 |
| MiniGPT-4 | 9.65 | 4.66 | 0.25 | 9.6 | 2.11 | 0.36 | 12.43 | 2.13 | 0.41 | 10.56 | 2.97 | 0.34 |
| LLaVA | 13.87 | 13.76 | 1.21 | 17.45 | 8.4 | 2.26 | 16.82 | 9.55 | 2.11 | 16.05 | 10.57 | 1.86 |
| BLIP2 | 15.03 | 19.31 | **2.78** | 14.88 | 5.66 | 0.84 | 14.33 | 5.77 | 0.97 | 14.75 | 10.25 | 1.53 |
| InstructBLIP | 12.61 | 13.69 | 1.11 | 14.46 | 2.81 | 2.15 | 15.55 | 2.45 | 1.79 | 14.21 | 6.32 | 1.68 |
| Otter | 14.08 | 20.15 | 1.31 | 14.93 | 5.41 | 0.68 | 15.21 | 4.72 | 0.72 | 14.74 | 10.09 | 0.9 |
| Cheetah | 12.13 | 13.35 | 0.67 | 13.22 | 2.73 | 0.24 | 12.45 | 2.21 | 0.24 | 12.6 | 6.1 | 0.38 |
| MMICL | 18.6 | 4.06 | 1.44 | 17.99 | 12.34 | 1.97 | 19.57 | 11.84 | 2.16 | 18.72 | 9.41 | 1.86 |
| GPT-4V | 12.47 | 19.94 | 0.92 | 19.11 | 10.64 | 1.87 | 18.96 | 11.99 | 2.05 | 16.85 | 14.19 | 1.61 |
| Gemini Pro | 12.97 | 20 | 1.6 | 18.84 | 6.53 | 0.83 | 17.73 | 4.36 | 0.91 | 16.51 | 10.3 | 1.11 |
| SAM | **20.93** | **20.19** | **2.78** | **19.4** | **19.92** | **3.27** | **20.36** | **18.35** | **3.62** | **20.23** | **19.49** | **3.22** |

(b) Storytelling

| Model | AESOP | | | VIST | | | DM800K | | | Average | | |
|---|---|---|---|---|---|---|---|---|---|---|---|---|
| | ROUGE-L | CIDEr | BLEU-4 | ROUGE-L | CIDEr | BLEU-4 | ROUGE-L | CIDEr | BLEU-4 | ROUGE-L | CIDEr | BLEU-4 |
| MiniGPT-4 | 15.42 | 2.29 | 0.53 | 10.36 | 1.4 | 0.17 | 12.01 | 0.89 | 0.49 | 12.6 | 1.53 | 0.4 |
| LLaVA | 14.52 | 2.92 | 0.98 | 8.66 | 0.22 | 0.3 | 11.35 | 0.35 | 0.97 | 11.51 | 1.16 | 0.75 |
| BLIP2 | 21.49 | 19.88 | 2.57 | 18.31 | 31.7 | 2.38 | 12.35 | 8.47 | 1.43 | 17.38 | 20.02 | 2.13 |
| InstructBLIP | 19.12 | 18.28 | 2.72 | 16.96 | 28.79 | 2.38 | 8.21 | 7.75 | 1.22 | 14.76 | 18.27 | 2.11 |
| Otter | 11.4 | 5.41 | 0.74 | 7.32 | 2.35 | 0 | 9.5 | 3.65 | 0.69 | 9.41 | 3.8 | 0.48 |
| Cheetah | 20.03 | 21.5 | 2.87 | 17.89 | 31.76 | 2.63 | 11.67 | 9.26 | 1.01 | 16.53 | 20.84 | 2.17 |
| MMICL | 15.2 | 13.04 | 1.39 | 12.13 | 2.87 | 0.68 | 11.79 | 7.4 | 0.82 | 13.04 | 7.77 | 0.96 |
| GPT-4V | 17.45 | 18.48 | 1.41 | 11.76 | 23.75 | 0.86 | 10.37 | 11.03 | 0.59 | 13.19 | 17.75 | 0.95 |
| Gemini Pro | 20.52 | 21.5 | 3.37 | 13.84 | 22.1 | 1.74 | 12.96 | 9.9 | 1.56 | 15.77 | 17.83 | 2.22 |
| SAM | **23.45** | **25.92** | **4.13** | **20.85** | **39.32** | **3.65** | **14.33** | **11.16** | **1.97** | **19.54** | **25.47** | **3.25** |

# 5 EXPERIMENTS

## 5.1 Experimental Settings

We implement SAM model in LAVIS library [22], building upon InstructBLIP-vicuna7b [9] architecture. We use the AdamW [30] optimizer with $\beta = (0.9, 0.999)$, and a learning rate of 2e-5 along with a weight decay of 0.05. We employ a cosine learning rate decay mechanism, along with a warm-up phase of 240 steps. We conduct training SAM using a batch size of 20 with 4 A100 GPUs.

**Datasets.** We conduct zero-shot evaluations on Group Captioning and Storytelling tasks to evaluate SAM's generalization ability. Both tasks demand semantic alignment across contextually different images. For the group captioning task, we select three datasets: Conceptual [25], Animal and Vehicle [13]. For the storytelling task, we select AESOP [38], VIST [15] and DM800K [6] as test sets.

**Metrics.** We measure the performance using a variety of captioning metrics. Following Forbes et al. [11], we report the scores on ROUGE-L [26], CIDEr [43] and BLEU-4 [34].

**Baselines.** For a comprehensive comparison, we consider open-source state-of-the-art MLLMs including MiniGPT-4 [50], LLaVA [28], BLIP2 [23], InstructBLIP [9], Otter [21], Cheetah [24] and MMICL [49], as well as industrial multi-modal chatbots including GPT-4V [32] and Gemini Pro [2].

Please refer to supplementary Section 3 for details about testing datasets, metrics and baselines.

## 5.2 Performance Comparison

The overall results of SAM and baselines are listed in Table 1. From it we can observe that SAM consistently achieves superiority across all datasets on all evaluation metrics. In particular, SAM outperforms other baselines on CIDEr score by a large margin of **5.3 (37%)** on group captioning and **4.63 (22%)** on storytelling, which demonstrates that SAM generates more accurate and comprehensive answers. Beyond this, we find that the improvement of our method mainly manifests in two aspects:

**1) Enhance the ability to follow instructions that require identifying multi-image correlations.** We observe that many MLLMs struggle to follow instructions for identifying correlations, such as interpreting storytelling tasks through image captions. This leads to their answers containing only some keywords rather than coherent story descriptions. We speculate that this issue stems from their training data lacking semantic correlations between images. For instance, MMICL relies on in-context learning data that might not be inherently related. In contrast, our training data emphasizes inter-image connections across contextually different images, enabling our model to excel at discerning associations and responding effectively to tasks that require identifying correlations, which further indicates the effectiveness of our training dataset.

**2) Enhance semantic alignment between contextually different images.** We observe that the state-of-the-arts often misinterpret the relations between images with diverse contexts. For instance, they may mix up character identities across images or

Table 2: Ablation study of different modules in SAM over group captioning and storytelling. $+\Delta_{\text{DATA}}$ means using our proposed data for training; $+\Delta_{\text{BSG}}$ means adding bidirectional semantic guidance without the adaptive adjustment module; $+\Delta_{\text{AA}}$ means adding adaptive adjustment module; $+\Delta_{\text{LIN}}$ means replacing the entire W-former with a simple linear layer.

(a) Group Captioning

| | $+\Delta_{\text{DATA}}$ | $+\Delta_{\text{BSG}}$ | $+\Delta_{\text{AA}}$ | $+\Delta_{\text{LIN}}$ | Group | | | Animal | | | Vehicle | | | Average | | |
|---|---|---|---|---|---|---|---|---|---|---|---|---|---|---|---|---|
| | | | | | ROUGE-L | CIDEr | BLEU-4 | ROUGE-L | CIDEr | BLEU-4 | ROUGE-L | CIDEr | BLEU-4 | ROUGE-L | CIDEr | BLEU-4 |
| 1 | | | | | 12.61 | 13.69 | 1.11 | 14.46 | 2.81 | 2.15 | 15.55 | 2.45 | 1.79 | 14.21 | 6.32 | 1.68 |
| 2 | ✓ | | | | 17.87 | 15.61 | 2.05 | 18.81 | 11.18 | 2.69 | 19.68 | 8.78 | 2.6 | 18.79 | 11.86 | 2.45 |
| 3 | ✓ | | | ✓ | 17.36 | 15.86 | 2.17 | 17.81 | 10.29 | 2.31 | 19.56 | 12.87 | 2.45 | 18.24 | 13.01 | 2.31 |
| 4 | ✓ | ✓ | | | 19.75 | 18.31 | 2.53 | 18.94 | 16.87 | 3.24 | 19.78 | 15.01 | 3.35 | 19.49 | 16.73 | 3.04 |
| 5 (ours) | ✓ | ✓ | ✓ | | 20.93 | 20.19 | 2.78 | 19.4 | 19.92 | 3.27 | 20.36 | 18.35 | 3.62 | 20.23 | 19.49 | 3.22 |

(b) Storytelling

| | $+\Delta_{\text{DATA}}$ | $+\Delta_{\text{BSG}}$ | $+\Delta_{\text{AA}}$ | $+\Delta_{\text{LIN}}$ | AESOP | | | VIST | | | DM800K | | | Average | | |
|---|---|---|---|---|---|---|---|---|---|---|---|---|---|---|---|---|
| | | | | | ROUGE-L | CIDEr | BLEU-4 | ROUGE-L | CIDEr | BLEU-4 | ROUGE-L | CIDEr | BLEU-4 | ROUGE-L | CIDEr | BLEU-4 |
| 1 | | | | | 19.12 | 18.28 | 2.72 | 16.96 | 28.79 | 2.38 | 8.21 | 7.75 | 1.22 | 14.76 | 18.27 | 2.11 |
| 2 | ✓ | | | | 22.4 | 23.77 | 3.74 | 19.64 | 35 | 2.81 | 13.88 | 8.73 | 1.8 | 18.64 | 22.5 | 2.78 |
| 3 | ✓ | | | ✓ | 18.94 | 21.71 | 3.24 | 14.08 | 15.55 | 0.77 | 13.97 | 3.74 | 1.24 | 15.66 | 13.67 | 1.75 |
| 4 | ✓ | ✓ | | | 23.14 | 24.35 | 3.83 | 20.19 | 36.84 | 3.48 | 13.98 | 10.73 | 1.68 | 19.1 | 23.97 | 3 |
| 5 (ours) | ✓ | ✓ | ✓ | | 23.45 | 25.92 | 4.13 | 20.85 | 39.32 | 3.65 | 14.33 | 11.16 | 1.97 | 19.54 | 25.47 | 3.25 |

Table 3: Performance comparison on change captioning tasks. The best is bolded and the second best is underlined.

| Model | Average | | |
|---|---|---|---|
| | ROUGE-L | CIDEr | BLEU-4 |
| MiniGPT-4 | 13.27 | 0.92 | 0.44 |
| LLaVA | 12.03 | 0.45 | 0.67 |
| BLIP2 | 13.44 | 2.77 | 0.34 |
| InstructBLIP | 10.9 | 1.87 | 0.5 |
| Otter | 12 | 2.36 | 0.35 |
| MMICL | 14.73 | 2.54 | 0.51 |
| GPT-4V | 17.12 | 3.38 | **1.2** |
| Gemini Pro | **17.59** | 2.87 | 1.05 |
| SAM | 16.08 | **3.84** | 1.08 |

Table 4: Inference efficiency test of InstructBLIP and SAM.

| Model | Average inference time | Average GPU memory usage |
|---|---|---|
| InstructBLIP | 4.9s | 17.12 GB |
| SAM | 5.1s (+4.1%) | 17.13 GB (+0.06%) |

invent details not present in the images. We attribute these hallucinatory responses to the absence of semantic alignment in their visual tokens. The existing MLLMs isolate the currently perceived image from its semantic contexts when extract its initial visual tokens, leading to semantic misalignment. Interestingly, similar issues also arise in industrial multi-modal chatbots such as GPT-4V and Gemini Pro, resulting in suboptimal performance. This indicates that these chatbots may not pay sufficient attention to multi-image semantic alignment. However, our approach, with the bidirectional semantic guidance mechanism, can mitigate the issues by accurately aligning the visual tokens of currently perceived image with its contexts using contextual semantics.

## 5.3 In-Depth Analysis

We further validate 5 vital issues of SAM as follows.

**1. Each component in SAM contributes to performance improvement.** We conduct an ablation study to illustrate the effectiveness of each component in Table 2. We start with the baseline model in Row 1 of the table that uses only the Q-former module to extract visual tokens for each image. **1)** We fine-tune the baseline model on our MmLINK dataset. The experiment results shown in Row 2 demonstrate that *our synthetic training data enhances*

*model's semantic alignment ability across contextual different images.* **2)** Next, we introduce the contextual semantics extracted by a simple linear layer to guide the visual token extraction process. The experiment results in Row 3 show that this straightforward approach brings either no improvement or even a decline in performance. We speculate that this simple method introduces uncertain contextual semantics with heavy noise, which misguides the visual token extraction process. Consequently, we adopt the bidirectional semantic guidance mechanism based on averaged contextual image features, which significantly improves the performance as shown in Row 4. It demonstrates that *bidirectional semantic guidance mechanism can accurately align visual tokens of currently perceived image with its contexts.* **3)** Finally, replacing average contextual image features with merged contextual features processed through our adaptive adjustment module leads to enhancement in all tasks, as indicated by the results in Row 5. This demonstrates that *the adaptive adjustment module effectively filters out irrelevant information from contextual images, thus improving patch-level alignment and the quality of bidirectional semantic guidance.*

**2. SAM still performs well when reasoning on highly similar images.** We conduct experiments on change captioning task, which requires model to describe subtle differences between highly similar images. We test on 4 datasets: IEdit [3], Spot-the-Diff [17], Birds-to-Words [11] and CLEVR-Change [35]. Dataset details are provided in supplementary Section 3.1. The average scores on the 4 datasets are reported in Table 3. From it, we find that compared to the baseline InstructBLIP, SAM does not show a performance decline. In contrary, SAM demonstrates superior performance comparing with those open-source state-of-the-art MLLMs, and even surpasses industrial multi-modal chatbots on CIDEr score. This indicates that introducing contextual semantics for alignment will

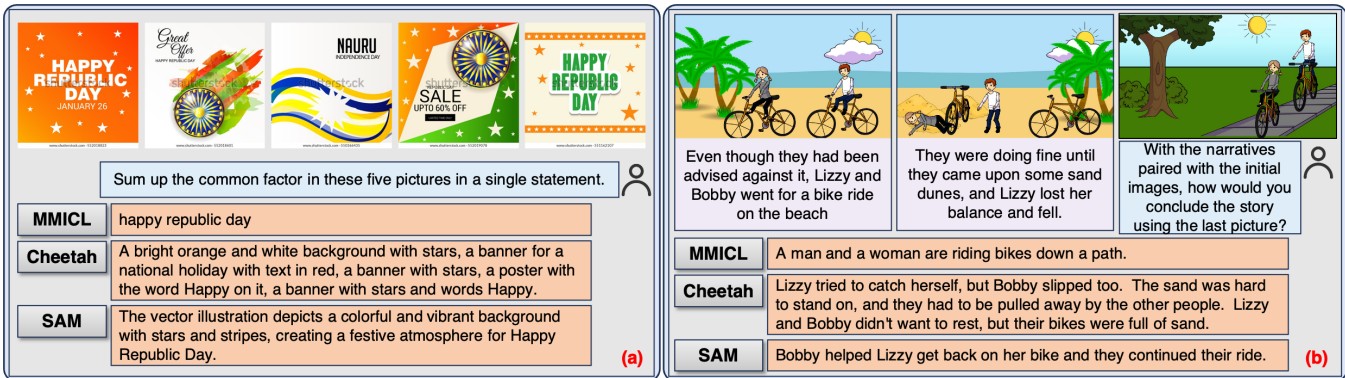

Figure 4: Case examples generated by SAM and other MLLMs. Other MLLMs' answers show either weak instruct-following ability or contain hallucinations, while SAM successfully performs semantic alignment and produces accurate responses.

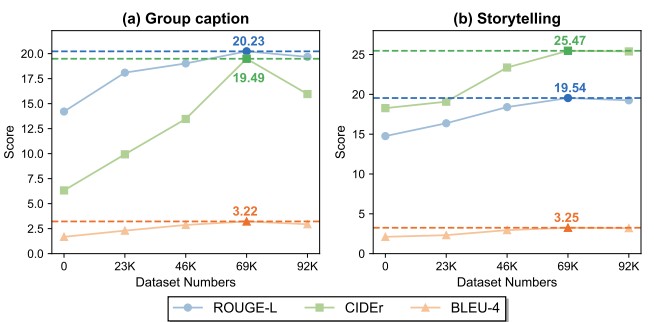

Figure 5: Performance of different data volumes.

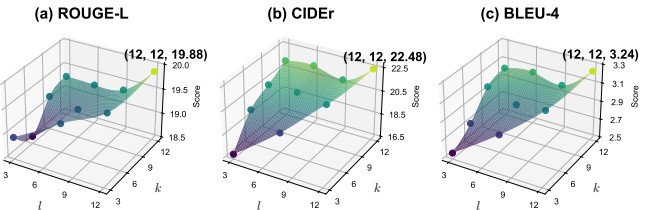

Figure 6: Average scores on 6 datasets of different interaction layers. $l$ is the layer that conveys initial visual tokens, $k$ is the layer that conveys contextual semantics, $k \geq l$ ($l$ and $k$ are defined in Section 4.2). The original 10 points are marked, and the surface are interpolated from these 10 points.

not degrade the model's reasoning performance on highly similar images; instead, it improves such capabilities.

**3. SAM does not significantly increase the inference costs.** We conduct inference efficiency test of SAM and the baseline, InstructBLIP. We report the average inference time and GPU memory usage of the two models generating a 20-tokens response in Table 4, which indicates that our proposed bidirectional semantic guidance mechanism does not bring obvious extra inference costs.

**4. Training SAM is data-efficient.** Our bidirectional semantic guidance mechanism effectively aligns visual tokens with the contexts, enhancing the model's ability to identify correlations, thus reducing the need for extensive training data. We further investigate the impact of different training data volumes. As shown in Figure 5, when the number of training data grows from 0 to 69K, the performance keeps increasing in group captioning and storytelling tasks. However, when the data volume reaches 92K, performance starts to decline. An additional increase in data volume leads to overfitting, confirming that training SAM is data-efficient.

**5. The final layers of the Q-former and the W-former are the best positions to conduct bidirectional semantic guidance.** We investigate the effect of different interaction layers within the Q-former and the W-former, which comprise 12 layers in total. We group every 3 layers into a step, ensuring that the interactive layer $k$, conveying contextual semantics, does not precede layer $l$, which conveys initial visual tokens ($l$ and $k$ are defined in Section 4.2). This approach yields 10 distinct configurations of interaction layers. Our findings, presented in Figure 6, highlight that early-stage

interactions do not enhance performance, whereas interactions in later layers lead to improvements. We speculate that this is because the initial layers of Q-formers capture less meaningful semantics, lacking the refined semantics needed for effective interaction. In contrast, the later layers provide high-level semantics for both Q-formers, significantly boosting performance.

## 5.4 Case Study

As illustrated in Figure 4, SAM demonstrates strong abilities to perform group captioning and storytelling tasks. In **(a)**, SAM can identify commonalities between images accurately, while other MLLMs' answers either show weak instruction-following ability or contain redundancy and hallucinations. In **(b)**, while other MLLMs might treat the storytelling task as an image captioning task, SAM successfully discovers the correlation between the characters in the images and matches them with the names of the characters in the text, creating a coherent story. More cases are illustrated in supplementary Section 4.

## 6 CONCLUSION

In this work, we introduce SAM, an innovative MLLM that utilizes bidirectional semantic guidance to enhance semantic alignment between contextual different images in multi-model instructions. As the test bed, we propose the MmLINK dataset. Experiments on MmLINK prove the effectiveness of our SAM model.

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
