# OpenReview forum: "Semantic Alignment for Multimodal Large Language Models"
_acmmm.org/ACMMM/2024/Conference — MM2024 Poster_

### Official Review · Reviewer_8jsS · 2024-05-21

**Rating:** 4
**Confidence:** 4

**Summary:**

This paper introduces Semantic Alignment for Multi-modal large language models (SAM), which is specifically optimized for multi-image cross-modal instructions. The SAM model introduces a Bidirectional Semantic Guidance Mechanism to enhance the preservation of linking information between different images and align their semantics before feeding them into the Large Language Model (LLM).

**Strengths:**

1. This paper proposes a large-scale dataset named MmLINK consisting of 69K samples and this dataset comprises multi-modal instructions with significantly diverse images.
2.  This paper has a commendable research motivation, aiming to establish semantic correlations between data with significant differences in input, thereby achieving semantic alignment between input images. Concretely, This paper proposes the multi-modal model named SAM, which employs bidirectional semantic guidance between images of the input multi-modal instruction to align the semantics of different images in the perception stage.
3. The SAM model demonstrates good performance across multiple tasks.

**Limitations:**

1. The network architecture of this paper lacks innovation, and the important module Q-former in A is identical to the Q-former in previous work BLIP-2. Meanwhile, the Adaptive Weights in W-former are generated solely by a simple attention mechanism.
2. This paper lacks comparisons with related fine-tuning work; comparing only with the original large model is insufficient to demonstrate its superiority.

**Suitability:**

3

---

### Official Review · Reviewer_E1nf · 2024-05-24

**Rating:** 5
**Confidence:** 3

**Summary:**

This paper aims to address the complex context issue in multi-image cross-modal instruction. It introduces a novel large-scale multi-modal dataset named MnLINK, and extensive experiments have conducted to demonstrate the effectiveness of the proposed method.

**Strengths:**

1. Simple and easy to understand
2. Clear motivation

**Limitations:**

1. Adaptive weights aim to eliminate patches that are irrelevant to the perceived images. However, the adjustment module is not perceived-image-aware, which raises concerns about its rationale and effectiveness.

2. Lacking training objectives

3. It is better to add a table comparing the proposed dataset with previous datasets

**Suitability:**

3

---

### Official Review · Reviewer_Cce4 · 2024-05-24

**Rating:** 4
**Confidence:** 3

**Summary:**

This paper introduces Semantic Alignment for Multi-modal large language models (SAM), that utilizes bidirectional semantic guidance to enhance semantic alignment between contextual different images in multi-model instructions. As the test bed, it proposes a large-scale dataset named MmLINK consisting of multi-modal instructions with significantly diverse images. The experiments on the group captioning task and the storytelling task prove the effectiveness of the SAM model.

**Strengths:**

The exploration of the semantic correlation between substantially diverse images in the multi-modal instruction within the MLLMs field is inspiring.

The proposed dataset MmLINK is interesting and novel.

The paper is well-organized and easy to follow.

**Limitations:**

1. In Figure 3, how are the vectors $w^l$ and $h^i$ combined with image features? What are their dimensions?
2. In Q-former, ${Query}$ is a concatenation of pre-trained query vectors and textual tokens. According to the transformer calculation criteria, its output dimension must be consistent with ${Query}$ , and its length is the total length of pre-trained query vectors and textual tokens. Why is the output length of Q-former equal to the total number of viusal tokens in all images? This is very strange and unreasonable.
3. Experiments show that the proposed model can generate better performance than existing multi-modal large models. It is well worth investigating whether the proposed technique can produce good results when applied to other backbones (such as LLaVA, BLIP2).

**Suitability:**

3

---

### Meta-Review · Area_Chair_n1XG · 2024-07-05

**Recommendation:** Accept (Poster)
**Confidence:** 4

**Metareview:**

This paper is a pioneering investigation into the ability of existing MLLMs to establish semantic correlations between significantly different images. All reviewers agree that the problem is well-motivated, and the proposed large-scale MmLINK dataset, which includes multi-modal instructions with significantly diverse images, is useful. While the proposed SAM model may have limited novelty, the AC believes that the contribution is sufficient and that the paper's quality meets the acceptance threshold.

It is strongly recommended that the authors incorporate the rebuttal discussion and reviewer comments into the final version of the paper to further enhance its quality.